# Obesity and Pancreatic Diseases: From Inflammation to Oncogenesis and the Impact of Weight Loss Interventions

**DOI:** 10.3390/nu17142310

**Published:** 2025-07-14

**Authors:** Mariana Souto, Tiago Cúrdia Gonçalves, José Cotter

**Affiliations:** 1Gastroenterology Department, Unidade Local de Saúde do Alto Ave, 4835-044 Guimarães, Portugal; tiagomcg@hotmail.com (T.C.G.); jabcotter@gmail.com (J.C.); 2Life and Health Sciences Research Institute (ICVS), School of Medicine, University of Minho, 4704-553 Braga, Portugal; 3ICVS/3B’s—PT Government Associate Laboratory, 4710-057 Braga, Portugal

**Keywords:** obesity, pancreas, acute pancreatitis, chronic pancreatitis, pancreatic cancer

## Abstract

**Background**: Obesity is a growing global health concern and a modifiable risk factor for multiple pancreatic diseases, including acute pancreatitis (AP), chronic pancreatitis (CP), and pancreatic cancer (PC). While these conditions have distinct clinical courses, obesity contributes to their pathogenesis through shared mechanisms, such as visceral adiposity, systemic inflammation, insulin resistance, and ectopic pancreatic fat deposition. **Methods**: This narrative review synthesizes current evidence from clinical, epidemiological, and mechanistic studies exploring the relationship between obesity and pancreatic diseases. We also critically evaluate the effects of weight loss interventions—including lifestyle modifications, pharmacologic therapies, endoscopic approaches, and bariatric surgery—on the risk and progression of disease. **Results**: Obesity increases the risk and severity of AP via mechanisms such as gallstone formation, hypertriglyceridemia, and lipotoxicity. In CP, obesity-related intrapancreatic fat and metabolic dysfunction may influence disease progression, although some data suggest a paradoxical protective effect. In PC, obesity accelerates tumorigenesis through chronic inflammation, adipokine imbalance, and activation of oncogenic signaling pathways. Weight loss interventions, particularly bariatric surgery and incretin-based therapies (e.g., GLP-1 receptor agonists and dual agonists such as tirzepatide), show promising effects in reducing disease burden and improving metabolic and inflammatory profiles relevant to pancreatic pathology. **Conclusions**: Obesity plays a multifaceted role in the pathophysiology of pancreatic diseases. Therapeutic strategies targeting weight loss may alter disease trajectories, improve outcomes, and reduce cancer risk. Further research is needed to define optimal intervention strategies and to identify and validate biomarkers for personalized risk assessment and prevention.

## 1. Introduction

Obesity has emerged as a critical global health issue, with far-reaching consequences beyond cardiovascular and metabolic diseases [1,2]. By 2050, it is projected that 3.8 billion adults—more than half of the expected global adult population—will be living with overweight or obesity [1]. Obesity is a significant risk factor for all-cause mortality [3]. Among these, the pancreas has gained increasing recognition as a metabolically sensitive organ profoundly influenced by excess adiposity [4]. Paralleling the obesity epidemic, the global burden of pancreatic diseases has risen markedly over the past decades [5,6].

Accumulating evidence suggests that obesity contributes to the entire spectrum of pancreatic disorders—including acute pancreatitis (AP), chronic pancreatitis (CP), and pancreatic cancer (PC)—through distinct yet interconnected pathophysiological mechanisms [7,8,9]. These include systemic and local inflammation, ectopic fat deposition, insulin resistance, and dysregulated adipokine signaling. Moreover, visceral and intrapancreatic fat play a pivotal role in modulating the onset, severity, and progression of these conditions, particularly through mechanisms involving lipotoxicity and altered immunometabolism [9].

Recent work has also underscored the importance of nutrition in shaping outcomes across pancreatic diseases, highlighting the need for systematic and individualized nutritional strategies from early in the disease course [10,11]. While previous studies have explored the impact of obesity on individual pancreatic diseases, a comprehensive synthesis integrating mechanistic insights with clinical and therapeutic implications remains limited. Furthermore, the rising use of obesity treatments—including lifestyle interventions, pharmacotherapy, endoscopic procedures, and bariatric surgery—has raised important questions about their potential to modify pancreatic disease risk and outcomes. This narrative review seeks to bridge existing knowledge gaps by critically examining the complex interplay between obesity and pancreatic pathology while evaluating the potential of weight loss interventions to mitigate disease burden. It presents a novel and integrated synthesis that connects fragmented evidence across disease stages, sheds light on underexplored immunometabolic mechanisms, and frames emerging anti-obesity therapies within the evolving landscape of pancreatic disease prevention and treatment.

## 2. Methods

This is a narrative review based on a comprehensive literature search conducted in PubMed. Articles published between January 2000 and May 2025 were considered. The selection focused on studies exploring the relationship between obesity and pancreatic diseases—namely AP, CP, and PC. We included clinical trials, observational studies, mechanistic research, relevant preclinical data, and selected review articles to provide a broader context and support conceptual integration across diverse research domains. Additional articles were identified through manual screening of reference lists from key reviews.

The inclusion criteria comprised English-language publications addressing obesity in the context of pancreatic pathology. The exclusion criteria were non-peer-reviewed articles and studies not directly relevant to the topic. Due to the broad scope and heterogeneity of the available evidence, no formal risk of bias assessment or quantitative synthesis was performed.

Instead, this review aimed to provide an integrative overview by connecting findings across epidemiology, molecular biology, clinical research, and therapeutic developments. We acknowledge the inherent limitations of the narrative review format, including potential selection bias and subjectivity in study inclusion.

## 3. The Role of Obesity in the Onset and Severity of Acute Pancreatitis

### 3.1. Obesity as a Risk Factor for Acute Pancreatitis

Obesity, particularly characterized by increased visceral adiposity, has been consistently associated with an elevated risk of developing AP. While the most common etiologies of AP include gallstones and alcohol abuse, obesity amplifies susceptibility to AP both directly and through its metabolic complications [12,13]. Multiple studies and meta-analyses have established a dose-dependent relationship between increased body mass index (BMI) and waist circumference (WC) and the risk of AP, with the risk increasing steeply beyond a BMI of 30 kg/m^2^ [14,15,16,17]. One meta-analysis indicated an 18% increased risk of AP per 5-unit BMI increment and a 36% risk increase per 10 cm increase in WC, underscoring the role of visceral fat over general adiposity [14].

### 3.2. Mechanisms Linking Obesity to the Onset of Acute Pancreatitis

#### 3.2.1. Gallstone Formation

First, obesity increases the prevalence of cholelithiasis, a leading cause of pancreatitis [18,19]. This is primarily due to cholesterol supersaturation in bile, gallbladder hypomotility, and changes in bile acid metabolism, all of which are common in obese individuals. As a result, the formation of gallstones becomes more frequent, increasing the likelihood of biliary obstruction and subsequent pancreatic inflammation [20]. The risk is particularly high in individuals with visceral obesity, which has a stronger correlation with gallstone-related pancreatitis than general obesity [18,21].

#### 3.2.2. Hypertriglyceridemia-Associated Pancreatitis

Obesity is frequently associated with secondary and unmasked primary hypertriglyceridemia (HTG), another well-established cause of AP [22,23]. There is still uncertainty surrounding the exact pathophysiological processes involved in HTG-AP. Among the potential mechanisms, elevated triglyceride levels promote the production of triglyceride-rich lipoproteins, which are broken down by pancreatic lipase during inflammation into free fatty acids (FFAs), particularly unsaturated fatty acids (UFAs) [24]. These FFAs can damage pancreatic acinar cells directly by inducing mitochondrial dysfunction, lowering ATP production, increasing intracellular calcium, and triggering necrosis [25]. Another theory focusing on the mechanisms of HTG-AP is increased blood viscosity in HTG, which reduces tissue microcirculation, possibly leading to ischemia and acidosis in pancreatic capillaries [24]. HTG is both a cause and a complication of AP in obese patients, creating a vicious metabolic cycle.

#### 3.2.3. Insulin Resistance and Type 2 Diabetes Mellitus

Obesity is a key driver of insulin resistance and type 2 diabetes mellitus (T2DM), both of which have been associated with an increased risk of AP [26,27,28]. T2DM may contribute to AP through mechanisms such as oxidative stress, inflammation (e.g., NF-κB and TNF-α activation), lipid peroxidation, and associated conditions like HTG and cholelithiasis [28,29,30]. Incretin-based therapies, including GLP-1 receptor agonists and DPP-4 inhibitors, have been linked to rare cases of drug-induced AP [31,32,33]. While some studies suggest an increased AP risk in diabetic patients or those on these medications, others find no significant association, and evidence remains inconclusive [34]. Additionally, data on AP outcomes in diabetic patients are conflicting, highlighting the need for further research [30,35,36].

#### 3.2.4. Intrapancreatic Fat as a Substrate for Inflammation

Fat infiltration of the pancreas itself—known as intrapancreatic fat (IPF)—additionally increases with BMI and may provide an abundant substrate for lipolysis during an acute insult [37,38,39]. This environment not only predisposes to the onset of AP but also lays the groundwork for a more severe inflammatory cascade once pancreatitis is initiated [39].

### 3.3. Obesity and the Severity of Acute Pancreatitis

Beyond its role in the development of AP, obesity significantly influences its clinical course and severity. Numerous meta-analyses report that obese individuals are at a higher risk of severe acute pancreatitis (SAP), local and systemic complications, organ failure, and mortality [17,40,41,42,43].

The pathophysiology underlying this increased severity is multifactorial. A key component is the interaction between digestive enzymes and visceral fat, especially within and around the pancreas. During AP, the leakage of lipases, such as pancreatic triglyceride lipase, into surrounding adipose tissue leads to lipolysis of triglyceride-rich adipocytes [44]. This results in the release of UFAs—primarily oleic and linoleic acid—which are directly toxic to pancreatic acinar cells [45]. These fatty acids disrupt mitochondrial function by inhibiting complexes I and V, causing ATP depletion and necrosis rather than apoptosis [39,44]. This necrotic pattern—especially peri-fat acinar necrosis—is predominant in obese patients and is a key marker of severity [39]. This cascade exemplifies lipotoxicity, a pathogenic process whereby excessive accumulation and breakdown of lipids—particularly in non-adipose tissues—result in cellular dysfunction and death. In the context of AP, lipotoxicity amplifies pancreatic necrosis and initiates a robust inflammatory response. UFAs act as potent inflammatory mediators by increasing levels of pro-inflammatory cytokines, such as TNF-α, CXCL1, and CXCL2, and triggering systemic inflammation [25]. This “lipolytic flux” not only worsens local pancreatic necrosis but also contributes to remote organ dysfunction, including acute kidney injury and acute respiratory distress syndrome, both of which are common complications in SAP [25,44].

### 3.4. Clinical Implications and Future Directions

The recognition of obesity as both a risk factor and modifier of disease severity in AP has direct clinical relevance. BMI and, more accurately, measures of visceral adiposity should be considered when stratifying patients for risk of SAP, organ failure, or local complications. This may support earlier admission to higher-level care or more aggressive monitoring protocols in obese individuals.

Nutritional support is a cornerstone in the management of AP, particularly in moderately severe and severe forms. The former concept of “pancreatic rest” has been replaced by the “gut rousing” theory. Oral or enteral nutrition, initiated early, does not exacerbate inflammation but instead helps preserve gut integrity and reduces complications such as infectious necrosis, organ failure, and mortality. In patients with mild AP, early reintroduction of a solid diet is safe and well-tolerated and may shorten the hospital stay without increasing symptom recurrence [11].

According to the recent European Society for Clinical Nutrition and Metabolism (ESPEN)/United European Gastroenterology (UEG) guidelines, despite the increased severity of AP in patients with obesity, there is no current indication for obesity-specific nutritional care during the acute phase beyond general recommendations, except in cases of severe HTG [46]. In these instances, fasting, intravenous fluid resuscitation, and subsequent dietary counseling—emphasizing caloric restriction and the replacement of saturated fats with mono- and polyunsaturated fat sources—are recommended foundational strategies [47].

The same guidelines further suggest that, in cases of SAP, an iso-caloric, high-protein diet (>1.3 g/kg adjusted body weight/day) may be warranted, particularly in the setting of elevated metabolic demand. Where feasible, indirect calorimetry is the preferred tool for guiding individualized energy and protein requirements [46].

Currently, there are no specific management guidelines for AP tailored to patients with overweight or obesity. However, given the pro-inflammatory state and metabolic demands of these patients, further research is warranted on tailored nutritional strategies.

From a research perspective, future studies should explore the utility of novel biomarkers, such as adipokines and inflammation-related lipid profiles, to improve risk prediction. Interventional trials targeting obesity-related mechanisms—such as oxidative stress or lipolytic flux—could yield new therapeutic strategies aimed at reducing AP severity or preventing recurrence.

## 4. Obesity and Chronic Pancreatitis: Risk, Mechanisms, and Clinical Perspectives

While obesity plays a clearly detrimental role in AP, its involvement in CP is less straightforward and may appear to confer paradoxical clinical outcomes. This section explores how pancreatic fat behaves differently in the chronic setting and the implications for disease progression.

### 4.1. Obesity and Risk of Chronic Pancreatitis

While the association between obesity and AP is well recognized, the relationship with CP remains more nuanced and less well defined. Evidence regarding the impact of obesity on the risk and progression of CP is both limited and sometimes contradictory. A meta-analysis suggested that a 5-unit increase in BMI may be associated with a 22% reduction in CP risk, although this was based on only two prospective studies and should be interpreted with caution due to potential reverse causality or pre-existing undiagnosed CP [14].

Some observational data suggest a potential protective role of excessive body weight in CP, with studies reporting reduced mortality and better clinical outcomes among patients with a BMI ≥ 25 kg/m^2^ [48]. Overweight individuals have shown improved islet yields during autologous islet transplantation, and children with obesity and CP were less likely to develop exocrine insufficiency or require pancreatectomy [49,50]. These findings have led to the hypothesis of an “obesity paradox” in CP. However, this concept remains speculative and may reflect methodological limitations, such as reverse causation (e.g., weight loss due to disease severity) or selection bias.

From a mechanistic perspective, adipocyte mass in CP is not directly correlated with BMI and is often embedded within fibrotic stroma. This fibrotic encapsulation may restrict lipolytic flux and limit the pro-inflammatory impact of local fat depots—contrasting with the acute lipotoxicity seen in AP [25,51]. Such structural remodeling may mitigate the harmful effects of IPF during chronic disease stages.

Additionally, a higher BMI may reflect greater metabolic reserve or lower sarcopenia risk rather than a direct protective role of adiposity. Obesity is heterogeneous in nature, varying by fat distribution, inflammatory profile, and muscle mass, and this can influence CP progression. For example, individuals with preserved lean mass and moderate adiposity may have different outcomes than those with sarcopenic or visceral obesity. Moreover, variability in IPF activity, adipokine profiles, and immune–metabolic responses may further shape disease trajectories.

These nuanced variables underscore the complexity of interpreting the relationship between obesity and CP. Current evidence does not robustly support a genuinely protective role of obesity in CP, and further well-designed, prospective studies are needed to unravel this paradox.

On the other hand, other studies have suggested that a hypercaloric diet—particularly one rich in fats and proteins—may contribute to the development of CP [52,53]. Furthermore, being overweight prior to the onset of disease has been identified as a potential risk factor, especially in cases of alcohol-related CP [53]. These observations indicate that, although obesity per se may not directly promote chronic fibrogenesis, the broader metabolic and nutritional environment associated with excess weight could modulate individual susceptibility, particularly in the context of dietary and alcohol-related risk factors.

### 4.2. Intrapancreatic Fat, Fibrosis, and Progression in Chronic Pancreatitis

Obesity is strongly associated with increased IPF, but its pathophysiological role in CP diverges markedly from that in AP. CP is now increasingly recognized as part of a disease continuum that often begins with episodes of AP, progresses through recurrent AP, and eventually evolves into chronic disease [54,55,56]. This progression involves morphological remodeling of the pancreas, characterized by loss of acinar tissue and its gradual replacement by fibrosis and adipose infiltration [25]. In CP, however, adipose tissue becomes compartmentalized within dense fibrotic septa that physically isolate fat from functional parenchyma [51]. This fibrotic architecture may attenuate the severity of acute flares by limiting the lipolytic injury seen in AP, where FFAs derived from peripancreatic fat exacerbate local necrosis [25].

Interestingly, the correlation between BMI and IPF appears to weaken in CP. Patients with CP often exhibit disproportionately high IPF compared to BMI-matched controls, suggesting disease-specific mechanisms of fat accumulation and remodeling [51]. Both adipocyte infiltration and intracellular lipid accumulation have been implicated in lipotoxic damage, oxidative stress, and paracrine signaling that further impair pancreatic function [9,57].

Although many observational studies have suggested an association between IPF and CP [58,59,60], recent genetic evidence supports a potential causal relationship. A Mendelian randomization study by Yamazaki et al., using genome-wide association data from over 25,000 individuals, demonstrated that genetically predicted IPF is significantly associated with both AP (odds ratio [OR] per 1-SD increase: 1.40; 95% CI: 1.12–1.76; *p* = 0.0032) and CP (OR: 1.64; 95% CI: 1.13–2.39; *p* = 0.0097). These results not only support the active role of IPF in the pathophysiology of pancreatitis but also suggest that reducing IPF may represent a promising preventive or therapeutic approach [61].

### 4.3. Clinical Implications and Future Directions

In the assessment of CP, clinicians should evaluate not only BMI but also fat distribution and the presence of metabolic comorbidities, as these factors may influence disease risk, progression, and response to therapy. Current ESPEN guidelines recommend a comprehensive nutritional assessment that includes clinical symptoms, organ function, anthropometric parameters (e.g., BMI and WC), and relevant biochemical markers. BMI alone is insufficient, particularly in obese patients, as it does not detect sarcopenia—an important determinant of functional status and prognosis in CP [46,62].

Although malnutrition remains common in advanced CP, a substantial proportion of patients present with overweight or obesity [63]. According to ESPEN/UEG guidelines, weight reduction should be encouraged in this subgroup, particularly in cases of CP unrelated to alcohol or tobacco use, where sarcopenia and micronutrient deficiencies are less prevalent [46]. However, weight loss strategies must be carefully tailored to avoid exacerbating exocrine insufficiency or nutritional deficiencies.

While fibrosis may offer some protection against acute necrotic flares, excessive IPF may still contribute to progressive dysfunction. Pancreatic enzyme replacement therapy (PERT) should be initiated in cases of suspected pancreatic insufficiency, with starting doses of 25,000 lipase units per meal and adjusted as needed [46].

Beyond its impact on pancreatic function and metabolic health, CP—particularly when influenced by obesity-related mechanisms—may contribute to neoplastic transformation. The combination of chronic inflammation, fibrosis, and IPF accumulation creates a microenvironment conducive to acinar-to-ductal metaplasia, a proposed early event in pancreatic carcinogenesis [9]. As such, understanding the interplay between obesity, CP, and PC is crucial. This potential progression will be addressed in the following section, which explores the relationship between obesity and PC.

Future research should explore the clinical utility of IPF quantification via imaging as a non-invasive biomarker to predict progression from AP to CP. Further research should clarify whether IPF reduction, via lifestyle or metabolic interventions, can alter the natural history of CP or improve clinical outcomes.

## 5. Obesity and Pancreatic Cancer: Epidemiology, Pathogenesis, and Therapeutic Perspectives

### 5.1. Epidemiology and Obesity as a Risk Factor

Globally, PC is the 12th most common malignancy and the 6th leading cause of cancer-related mortality [64]. The 5-year survival for patients with PC in the United States is about 12%, and PC is projected to become the second leading cause of cancer mortality by 2040, following lung cancer and surpassing colorectal cancer [65,66]. Pancreatic ductal adenocarcinoma (PDAC) is the most prevalent form of PC and represents one of the deadliest solid malignancies, largely due to its late diagnosis and aggressive progression [67].

Obesity is considered a modifiable risk factor for PDAC, acting independently or synergistically with other factors such as diabetes, CP, tobacco smoking, excessive alcohol intake, a diet rich in red and processed meat, and viral infections [68,69,70,71,72,73,74].

The incidence and mortality of PC are increasing worldwide, with obesity increasingly recognized as a contributing factor. The prevalence of obesity has reached pandemic proportions, paralleling the rising rates of PDAC, suggesting a potential epidemiological correlation [5,75].

Multiple large-scale epidemiological studies and meta-analyses have consistently demonstrated a positive correlation between obesity and PC [76,77,78,79,80]. A 5-unit increase in BMI is associated with an 8–12% rise in PC risk, with abdominal obesity also showing strong associations, particularly in women [77,78,79,80]. Studies have also found that overweight or obese individuals tend to develop PC at a younger age [81].

### 5.2. Mechanisms Linking Obesity to Pancreatic Cancer Development

Obesity contributes to PC through several interconnected biological processes. Chronic low-grade inflammation driven by adipose tissue expansion increases cytokines like IL-6, TNF-α, and IL-1β, promoting acinar cell injury and neoplastic transformation [9]. Adipokines, such as leptin and resistin, stimulate tumor growth, while reduced adiponectin removes protective anti-inflammatory effects [82,83].

Obese individuals often exhibit elevated insulin and insulin-like growth factor 1 (IGF-1) levels, which activate mitogenic pathways (e.g., Ras/ERK and PI3K/AKT/mTOR), leading to increased cellular proliferation and decreased apoptosis [84]. Obesity is also linked to pancreatic fat infiltration replacing acinar cells, which leads to structural changes, leading to reduced exocrine function and structural alterations that predispose to acinar-to-ductal metaplasia, a critical early step in pancreatic intraepithelial neoplasm formation [9]. Acinar-to-ductal metaplasia is exacerbated by KRAS mutations, which are present in over 90% of PDAC [9].

Obesity induces ectopic expression of cholecystokinin in pancreatic β-cells, which can independently drive KRAS-mediated tumorigenesis [85]. Additionally, altered gut microbiota in obesity may further promote tumor progression [86].

Metabolic reprogramming in obese environments boosts lipogenesis and cholesterol synthesis in cancer cells, while lipid accumulation triggers cellular stress and modifies the tumor microenvironment, favoring progression and resistance to therapy [9].

### 5.3. Clinical Implications and Future Directions

The increasing recognition of obesity as a contributor to pancreatic carcinogenesis has important implications for clinical practice and public health. Obese patients may be at increased risk of earlier onset and more aggressive PC phenotypes, highlighting the potential of metabolic and inflammatory markers associated with obesity as tools for early risk stratification and disease prediction [81].

Although body composition is not yet integrated into routine prognostic models for PC, emerging evidence suggests that premorbid obesity may negatively impact clinical outcomes, long-term survival, and, potentially, treatment efficacy [87,88]. For example, sarcopenic obesity—a condition characterized by excess fat and reduced muscle mass—has been associated with higher postoperative complication rates and poorer overall survival [88].

Surgical studies are mixed, with some reporting comparable outcomes across BMI categories while others note increased technical difficulty, blood loss, and perineural invasion in obese patients [89,90]. Regarding chemotherapy, obesity-induced inflammation and desmoplasia have been linked to reduced response to treatment in PDAC [91]. In contrast, data on radiotherapy outcomes remain limited, underscoring a need for further investigation into how obesity influences treatment efficacy [89,92]. These observations suggest that evaluating and addressing body composition could improve patient selection and optimize treatment strategies.

Parallel to the impact of obesity, nutritional deterioration—including cancer cachexia, malnutrition, and pancreatic exocrine insufficiency—is highly prevalent in PC and profoundly affects therapeutic tolerance, quality of life, and survival. Therefore, early and systematic assessment of nutritional status is essential and should be integrated into standard oncologic care. Guidelines recommend structured nutritional screening from the time of diagnosis, ideally involving multidisciplinary teams with specific expertise in clinical nutrition [10,11,93]. Simple clinical tools can support the early identification of patients at nutritional risk and help guide timely interventions. A recent retrospective study demonstrated that the Geriatric Nutritional Risk Index (GNRI), calculated using serum albumin and body weight, holds prognostic value in PC. Lower baseline scores and significant declines over time were independently associated with reduced overall survival, suggesting that the GNRI may be a practical and accessible method for longitudinal nutritional monitoring and risk stratification [94].

In PC, nutritional care must prioritize the mitigation of cachexia and maintenance of functional status. Once malnutrition is identified, appropriate interventions should be initiated promptly. Nutritional strategies—including individualized counseling, oral nutritional supplements, PERT, and, when necessary, enteral or parenteral nutrition—have been shown to mitigate nutritional decline and may improve survival outcomes in PC patients [10,11]. Obesity may coexist with malnutrition, underscoring the need for individualized dietary plans.

Metabolic interventions may offer adjunctive benefits. Metformin, a cornerstone in T2DM management, has shown promise in reducing PC risk and improving outcomes [95,96,97] by lowering insulin and IGF-1 levels and activating AMP-activated protein kinase pathways, thereby inhibiting cancer cell growth [98].

In summary, both obesity and nutritional decline compromise treatment response and long-term outcomes in PC. Integrating systematic assessment of body composition, nutritional status, and metabolic function into routine clinical workflows may help personalize care and improve prognoses. Furthermore, interventions aimed at weight reduction—whether lifestyle-based, pharmacologic, or surgical—warrant exploration both for cancer prevention and for their potential to enhance response to standard therapies.

Finally, additional research is needed to evaluate whether intentional weight loss can improve survival and therapeutic efficacy in patients with established PC. Identifying robust biomarkers of obesity-driven tumorigenesis, including gut microbiota and inflammatory profiles, may also enhance early detection and enable more targeted preventive approaches.

## 6. The Impact of Obesity Treatment on Pancreatic Diseases

Given the multifaceted role of obesity in the development and progression of pancreatic disorders—including AP, CP, and PC—there is growing interest in the potential for obesity treatments to alter the natural course of these diseases. Interventions such as lifestyle modifications, pharmacological weight loss therapies, endoscopic therapies, and bariatric surgery not only reduce body weight but also influence key metabolic and inflammatory pathways implicated in pancreatic pathology. However, the therapeutic impact of these strategies may vary significantly according to patient characteristics (e.g., metabolic profile, comorbidities), disease stage (acute vs. chronic vs. neoplastic), and the type of intervention employed. These variables must be considered when interpreting clinical outcomes and making therapeutic decisions.

This section explores current evidence regarding the impact of obesity treatment on various pancreatic diseases as well as emerging therapeutic opportunities.

### 6.1. Lifestyle Modification and Preventive Strategies

Lifestyle interventions—centered on dietary improvement, physical activity, and behavioral counseling—remain foundational in obesity management and may significantly influence pancreatic health. Caloric restriction and Mediterranean-style diets reduce systemic inflammation, improve lipid profiles, and reduce ectopic fat deposition, including in the pancreas [99,100,101,102].

Even modest weight loss is associated with a significant decline in visceral abdominal fat, lipid content in the pancreas with favorable changes in insulin resistance, and lipid metabolism, potentially reducing the risk of HTG-related AP and slowing CP progression [103]. Early intervention in patients with metabolic syndrome may be especially effective in preventing pancreatic complications before irreversible changes develop [13].

Importantly, lifestyle modifications—particularly structured programs involving diet and exercise—not only support weight reduction but also enhance pancreatic endocrine function. This effect may significantly reduce the risk of developing T2DM, especially among obese, insulin-resistant older adults [104].

Preclinical studies in both murine models and human tissues of PDAC have demonstrated that caloric restriction significantly reduces the expression of inflammation-related genes and suppresses tumor growth [105]. Furthermore, it has also been suggested that calorie restriction delays the progression of pre-PDAC lesions [106].

Despite the slower and more variable nature of lifestyle interventions compared to surgical or pharmacologic options, their accessibility and safety profile make them an essential component of long-term disease management and prevention.

### 6.2. Pharmacologic Interventions

Orlistat is a gastrointestinal lipase inhibitor that decreases fat absorption through the inhibition of pancreas and stomach lipases, resulting in a calorie intake reduction, approved by both the EMA and FDA (1998 and 1999, respectively) for the treatment of obesity [107].

The pancreatic safety of orlistat remains a concern due to reports of AP. While clinical trials have not established a causal relationship, rare instances of AP shortly after treatment initiation suggest possible idiosyncratic reactions [108,109,110]. Recent FDA adverse event reporting system-based pharmacovigilance analysis identified 21,079 adverse event reports involving orlistat, with signals such as elevated pancreatic enzymes (n = 8) and edematous pancreatitis (n = 3), reinforcing the need for caution. Although causality is unproven, these findings highlight the importance of clinical vigilance in high-risk individuals, given the potentially severe outcomes [111]. The underlying mechanism remains uncertain but may involve direct toxicity or hypersensitivity [109].

Beyond its role in weight loss, orlistat has also shown anti-cancer activity in laboratory studies. In pancreatic cancer cells (PANC-1), it reduced cell growth and increased cell death by blocking fatty acid synthase, an enzyme linked to tumor growth [112]. These findings suggest that orlistat or its optimized analogs may hold experimental promise as future therapeutic agents in PC. However, these results remain preclinical, and their translational relevance is uncertain. To date, no clinical trials have evaluated orlistat’s anti-cancer efficacy in PC patients.

GLP-1 receptor agonists (GLP-1 RAs) are incretin-based therapies widely used for the management of T2DM and obesity. These agents mimic the action of endogenous glucagon-like peptide-1, enhancing glucose-dependent insulin secretion, suppressing glucagon, delaying gastric emptying, and promoting satiety. As a result, GLP-1 RAs produce clinically meaningful weight loss, improved glycemic control, and cardiovascular benefits [113].

There have been concerns about a potential association between GLP-1 receptor agonists and the occurrence of AP, with some suggestions that they may also increase the risk of PC following the introduction of this pharmacological therapy [114]. However, multiple large-scale meta-analyses and randomized controlled trials (RCTs) have since addressed these concerns [115,116,117]. The LEADER and SUSTAIN-6 trials did not demonstrate a significant increase in pancreatitis incidence among patients using liraglutide or semaglutide compared to a placebo [118,119].

The potential association between GLP-1 RAs and PC remains an area of active investigation. While some studies have raised concerns due to the expression of GLP-1 receptors in pancreatic tissue, current clinical evidence does not support an increased risk [114,120,121]. In contrast, several in vitro and animal studies have demonstrated potential anti-tumorigenic effects of GLP-1 RAs through modulation of key metabolic and inflammatory pathways, including inhibition of NF-κB and PI3K/AKT/mTOR signaling [122,123,124]. Notably, liraglutide has been shown to reduce tumor proliferation and improve chemosensitivity in gemcitabine-resistant PC models [125,126]. Nonetheless, these effects have not been replicated in human clinical trials, and the evidence remains preliminary. The lack of robust translational data limits conclusions about their efficacy in PC management. However, in rodent studies, chronic GLP-1 receptor stimulation has been associated with *C*-cell hyperplasia and the development of medullary thyroid carcinoma (MTC), leading to formal contraindications in patients with multiple endocrine neoplasia type 2 (MEN2) or a family history of MTC [127]. Importantly, the translational relevance of these findings to human thyroid tissue is debated [128].

The current stance of the regulatory agencies EMA and FDA is that a causal link between incretin-based drugs and pancreatitis or PC has not been conclusively established [34].

Tirzepatide, a dual glucose-dependent insulinotropic polypeptide (GIP) and GLP-1 receptor agonist, has emerged as a powerful agent for obesity and T2DM. In clinical trials, such as SURMOUNT-1 and SURPASS, it demonstrated superior weight loss compared to other existing therapies [129,130,131,132]. A systematic review and meta-analysis on the safety issues of tirzepatide suggested that it was not associated with a significant risk of pancreatitis in patients with T2DM and obesity [133]. A recent systematic review and meta-analysis including over 14,000 participants across 17 RCTs confirmed that tirzepatide was not associated with an increased risk of confirmed clinical pancreatitis compared to a placebo, insulin, or GLP-1 RAs. Furthermore, tirzepatide significantly reduced fasting insulin, *C*-peptide, glucagon, and HOMA2-IR, indicating positive effects on β-cell function and insulin resistance—metabolic pathways that are highly relevant to pancreatic disease progression and oncogenesis [134].

Concerns about its dual GIP and GLP-1 receptor agonism remain theoretical, largely based on mechanistic data suggesting activation of pathways such as PI3K/AKT/mTOR, which are involved in cell proliferation [135]. However, these molecular effects have not been linked to adverse pancreatic outcomes in human studies to date [136]. Preclinical findings of *C*-cell hyperplasia and MTC in rodents have prompted contraindications in patients with MEN2 or a family history of MTC, although these effects appear species-specific due to low GLP-1 receptor expression in human thyroid tissue [135]. Continued long-term surveillance is warranted, but current data indicate a favorable pancreatic safety profile for tirzepatide.

### 6.3. Endoscopic Bariatric and Metabolic Therapies

The American Society for Gastrointestinal Endoscopy (ASGE) and the European Society of Gastrointestinal Endoscopy (ESGE) suggest the use of endoscopic bariatric and metabolic therapies plus lifestyle modification in patients with a BMI ≥ 30 kg/m^2^ or with a BMI of 27.0–29.9 kg/m^2^ with at least 1 obesity-related comorbidity [137].

The intragastric balloon (IGB) is a non-surgical, endoscopic weight loss intervention that induces early satiety and reduces gastric volume [138]. It is considered for patients who are not candidates for surgery, who prefer less invasive options, or who require a bridge to surgery due to severe obesity [139,140]. IGB has been shown to result in significant short- to medium-term weight loss and improvements in metabolic parameters [141,142].

However, there have been rare but serious complications reported, including AP. The proposed mechanisms include mechanical compression of the pancreas by the balloon, duodenal obstruction, or migration [143]. Case reports and adverse event databases (e.g., FDA MAUDE) have documented instances of pancreatitis, often resolving after balloon removal [144,145,146]. While the risk appears low, clinicians should be aware of it, particularly in patients presenting with abdominal pain post-procedure. Currently, there is no evidence suggesting that IGB influences the long-term risk of PC.

The Duodenal-Jejunal Bypass Liner (DJBL) is an endoscopic device that mimics bariatric surgery by preventing contact between food and the proximal small intestine, showing benefits in weight loss and glycemic control. However, it has been linked to pancreatic complications such as AP due to factors like device migration or ampulla of Vater compression [147,148]. Awareness of these risks is essential. The ASGE–ESGE guidelines recommend DJBL with lifestyle modification for obesity and T2DM, while emphasizing the need for careful monitoring due to potential adverse events [137].

### 6.4. Bariatric and Metabolic Surgery

Bariatric surgery remains an effective long-term intervention for sustained weight loss in individuals with severe obesity and has demonstrated systemic benefits beyond weight reduction [149,150]. There is growing evidence that bariatric surgery—including Roux-en-Y gastric bypass (RYGB) and sleeve gastrectomy—may influence the onset, progression, and outcomes of pancreatic diseases.

In the context of AP, short-term postoperative risk must be considered. Although rare, AP occurs in 0.2–1% of patients after laparoscopic RYGB, even in the absence of cholelithiasis [151,152]. Over the first three years after surgery, gallstones and pancreatitis account for 5% and 10% of all complications, respectively [153]. The primary mechanisms underlying this increased risk include rapid weight loss and gallbladder stasis [151]. International guidelines recommend ursodeoxycholic acid after bariatric surgery during rapid weight loss to prevent gallstone-related complications [46,154,155].

Despite this risk, recent evidence suggests that AP in patients with prior bariatric surgery tends to be less severe [156]. A large observational study showed significantly lower mortality, organ failure, intensive care unit admission, and resource use in these patients compared to non-surgical controls, with even better outcomes in biliary AP cases [157].

Recent studies have also highlighted the role of bariatric surgery in the management of HTG-AP. In a prospective 12-month follow-up study, patients undergoing laparoscopic sleeve gastrectomy experienced normalization of triglyceride levels and no recurrence of AP compared to a 47.4% recurrence rate in the conservatively managed group [158]. A 5-year follow-up study further confirmed that both laparoscopic sleeve gastrectomy and RYGB significantly reduced recurrence rates (15.4% and 11.1%, respectively) versus 57.1% with standard medical treatment [159]. These findings support the long-term efficacy of metabolic surgery in reducing recurrence and improving metabolic control in obesity-related HTG-AP. However, these studies were limited by small sample sizes and single-center designs.

Regarding CP, the impact of bariatric surgery remains less well defined. Emerging evidence suggests that metabolic bariatric surgery significantly reduces IPF, with meta-analyses demonstrating relative reductions of up to 35.9% following surgery [160]. Given that increased IPF has been identified as a potential risk factor for subclinical CP in observational cohorts [59] and supported as a causal factor in both AP and CP via Mendelian randomization analyses [61], it is plausible that reducing IPF may influence disease trajectory. However, most supporting data are derived from small-scale or preclinical studies, and direct evidence linking bariatric surgery to improved clinical outcomes in early or subclinical CP remains limited. Future prospective, multicenter studies are required to validate these findings and to better define which patient subgroups may derive the greatest benefit. Moreover, the anti-inflammatory milieu established post-surgery may attenuate ongoing fibrotic remodeling and preserve residual exocrine and endocrine function. Animal models show reduced fibro-inflammatory lesions post-surgery [161], and human studies report significant decreases in pancreatic steatosis and improvements in metabolic markers [162]. Nonetheless, these findings remain preliminary, and further prospective, long-term studies are needed to validate their clinical implications.

Bariatric surgery has also demonstrated protective effects against obesity-associated malignancies [163,164,165]. Retrospective cohort studies report significantly reduced PC incidence in patients undergoing bariatric surgery, with hazard ratios of 0.46 when compared to matched non-surgical controls [166,167]. Furthermore, a recent meta-analysis, including more than 3.7 million adults, found that metabolic bariatric surgery reduced the risk of developing PC by 54% [168]. Although some findings remain inconclusive—potentially due to variations in surgical techniques, follow-up duration, or patient populations—animal models have also confirmed reduced tumor formation and improved survival following bariatric surgery [169]. These metabolic interventions likely exert their benefits through modulation of systemic inflammation, insulin resistance, lipid metabolism, and pancreatic fat infiltration.

## 7. Conclusions

Obesity exerts a multifactorial influence on the pathogenesis, progression, and prognosis of pancreatic diseases. From amplifying systemic inflammation and metabolic dysfunction in AP to modulating fibrotic remodeling in CP and promoting oncogenic transformation in PC, excess adiposity has emerged as both a modifiable risk factor and a therapeutic target. Importantly, growing evidence supports the notion that weight loss achieved through dietary, pharmacological, endoscopic, or surgical strategies not only improves metabolic parameters but may also favorably alter the natural history of pancreatic disease. Given this, systematic assessment of obesity and nutritional status should be integrated into the diagnostic workup and longitudinal care of all patients with pancreatic pathology. Tailoring interventions to the stage of disease and individual metabolic profile—particularly through structured dietary counseling and personalized nutritional plans—has the potential to enhance outcomes and improve quality of life.

This review underscores key emerging concepts, such as the pathophysiological relevance of IPF, the interface between obesity and immunometabolism in pancreatic pathology, and the therapeutic potential of modern weight loss agents, including GLP-1 receptor agonists and dual incretin therapies. These agents represent a promising avenue but require further validation. RCTs are urgently needed to determine the safety and efficacy of these pharmacologic interventions—especially in patients with early or subclinical pancreatic disease. Such trials should identify which patients benefit most, and at what stage of disease and under which clinical conditions these therapies should be employed. This evidence will be crucial for informing future clinical guidelines that specifically address pancreatic disease management in obese patients.

Furthermore, translational research should aim to elucidate how obesity-modifying strategies influence key biological processes, such as inflammation, fibrosis, and tumorigenesis. Multiomic approaches—including lipidomic, transcriptomic, and microbiome analyses—may uncover molecular subtypes of obesity-associated pancreatic disease potentially responsive to targeted treatments. The integration of artificial intelligence, driven prediction tools, and personalized nutritional interventions could further advance early detection and stratified prevention strategies.

As the obesity epidemic continues to escalate globally, implementing proactive, evidence-based, and individualized strategies—particularly those centered on nutrition—will be essential to reduce the burden of pancreatic diseases linked to excess adiposity.

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
