# Peer review of "Obesity and Pancreatic Diseases: From Inflammation to Oncogenesis and the Impact of Weight Loss Interventions"

_nutrients, 2025, doi:10.3390/nu17142310_

Round 1
Reviewer 1 Report
Comments and Suggestions for Authors
Summary: The review paper examines the role that obesity and overweight play in acute and chronic pancreatitis, and, to a lesser degree, pancreatic cancer. It is written well and annotated with relevant references. The paper is broken down into readily understandable sections that appear to flow well and provide an overview of the topic. At the end of the report there is a brief discussion of the role of bariatric surgery and GLP-1 receptor agonists in the control of weight and their influence on pancreatic disease. Although the paper is not earth shattering, it does add at least incrementally to our current body of knowledge and is worthy of consideration for publication.
Author Response
Comment 1: "The review paper examines the role that obesity and overweight play in acute and chronic pancreatitis, and, to a lesser degree, pancreatic cancer. It is written well and annotated with relevant references. The paper is broken down into readily understandable sections that appear to flow well and provide an overview of the topic. At the end of the report there is a brief discussion of the role of bariatric surgery and GLP-1 receptor agonists in the control of weight and their influence on pancreatic disease. Although the paper is not earth shattering, it does add at least incrementally to our current body of knowledge and is worthy of consideration for publication. " Response 1: We sincerely thank the reviewer for the thoughtful and constructive feedback provided on our manuscript entitled "Obesity and pancreatic diseases: from inflammation to oncogenesis and the impact of weight loss interventions." We are grateful for the positive comments regarding the quality of writing, the organization and clarity of the sections, and the relevance of the references cited. We appreciate the reviewer’s observation that the manuscript adds incrementally to the current body of knowledge, particularly through its integration of evidence across the spectrum of pancreatic diseases. Although the review is narrative in nature, our intention was to provide a comprehensive and clinically relevant synthesis of mechanistic, epidemiological, and therapeutic data to support both researchers and clinicians in understanding the multifaceted impact of obesity on pancreatic pathology. Once again, we thank the reviewer for the careful evaluation of the manuscript.Reviewer 2 Report
Comments and Suggestions for Authors
The manuscript is a narrative review, and while it integrates known data, it does not clearly state what new insights it brings compared to prior reviews on obesity and pancreatic disease.
The abstract is generally well-structured and informative; however, it would benefit from improved punctuation consistency, refined grammar in some sentences, moderation of certain overstatements (e.g., "central role"), and clarification of terminology, particularly regarding specific therapies like tirzepatide, to enhance clarity, precision, and scientific rigor.
The associations between obesity, acute/chronic pancreatitis, and pancreatic cancer have been previously discussed in many reviews. This paper may lack sufficient originality unless it emphasizes new therapeutic or molecular pathways, which are not clearly highlighted.
There is no systematic methodology for article inclusion. The authors do not detail: search databases used (e.g., PubMed, Embase); inclusion/exclusion criteria; time period covered; risk of bias assessment. There are no quantitative analyses or novel insights derived from pooled data.
Some therapeutic claims (e.g., anti-cancer effects of orlistat or GLP-1 RAs) are heavily based on preclinical findings, which are presented with insufficient critical appraisal of their translational limits.
The discussion of the “obesity paradox” in chronic pancreatitis is not critically dissected, some conflicting data are mentioned, but no mechanistic hypotheses or explanations are elaborated to resolve the paradox.
The manuscript provides a comprehensive overview of the relationship between obesity and pancreatic diseases. However, I recommend several improvements to enhance its scientific value. First, the section would benefit from deeper critical analysis of the cited literature, particularly in distinguishing between preclinical and clinical findings, and weighing the strength of evidence. Second, the discussion occasionally overgeneralizes outcomes (e.g., weight loss interventions), without specifying context or patient subgroups. Additionally, while individual disease mechanisms are well presented, greater integration across conditions, such as how obesity may influence the progression from acute to chronic pancreatitis to pancreatic cancer, would strengthen the narrative. The section also lacks an explicit discussion of the limitations of the review itself, including the use of a narrative format and the absence of a systematic methodology. Lastly, several concepts (e.g., lipotoxicity, intrapancreatic fat, GLP-1 therapies) are repeated throughout without deeper synthesis. Addressing these issues would enhance the clarity, depth, and critical rigor of the discussion.
The should acknowledge the lack of a systematic search strategy, possible selection bias, and the absence of formal quality assessment of the cited studies. Moreover, the reliance on preclinical data and observational studies should be discussed, particularly where mechanistic extrapolations, or therapeutic implications are drawn from animal models or small cohorts.
The future research directions are mentioned, but they remain somewhat general. The manuscript could be strengthened by proposing more specific research questions or methodologies. For example:
- Can intrapancreatic fat quantification serve as a clinical biomarker for progression risk in pancreatitis?
- What are the effects of pharmacologic weight-loss therapies on pancreatic cancer outcomes in randomized trials?
- Are there molecular subtypes of obesity-driven pancreatic disease that would benefit from targeted interventions?
Additionally, discussion of multi-omic approaches, AI-based risk prediction, or personalized nutritional interventions would modernize the future outlook.
The review successfully outlines that weight loss may reduce disease burden; however, this is stated in general terms, without clear clinical guidance. To improve its utility, the authors could summarize:
- What types of interventions are most appropriate at different disease stages (e.g., lifestyle vs. bariatric surgery)?
- What are the potential risks and benefits of GLP-1 therapies in patients with subclinical pancreatic disease?
- Should obesity be systematically assessed in pancreatitis workup and surveillance?
The conclusion section is generic and doesn’t adequately summarize key clinical implications or concrete future research directions.
Comments on the Quality of English LanguageEnglish including grammar, style and syntax, should be improved through the professional help from English Editing Company for Scientific Writings.
Below are revealed some grammatical and lexical issues:
- Line 19: "on disease risk and progression" - should be: "on the risk and progression of disease".
- Line 30: "validate biomarkers for personalized risk assessment" - might read better as "identify and validate biomarkers".
- Line 166: "may even present paradoxical clinical outcomes" – should be: "may appear to confer paradoxical clinical outcomes".
- Line 329: "Metformin, a cornerstone in T2DM management have shown..." - should be: " Metformin, a cornerstone in T2DM management has shown...".
- Line 337: "exploration not only for cancer prevention but also for their potential..." – shoul be: "exploration both for cancer prevention and for their potential...".
Author Response
Comment 1: “The manuscript is a narrative review, and while it integrates known data, it does not clearly state what new insights it brings compared to prior reviews on obesity and pancreatic disease.”
Response 1: Thank you for this important observation. We agree that clarifying the novelty of our review is essential. Accordingly, we have revised the Introduction (page 1-2) and Conclusion (page 12-13) to explicitly state the key insights our review adds to the existing literature. These include:
- A focused synthesis of how obesity-related mechanisms (e.g., intrapancreatic fat, lipotoxicity, adipokines) intersect across AP, CP, and PC, highlighting shared and diverging pathways.
- A detailed discussion of emerging therapeutic implications, including GLP-1 receptor agonists, tirzepatide, bariatric interventions, and potential molecular targets.
- Integration of recent evidence (2023–2025), including genetic studies and preclinical findings not covered in previous reviews.
These additions clarify the distinct contribution of our work.
Comment 2: “The abstract is generally well-structured and informative; however, it would benefit from improved punctuation consistency, refined grammar in some sentences, moderation of certain overstatements (e.g., "central role"), and clarification of terminology, particularly regarding specific therapies like tirzepatide, to enhance clarity, precision, and scientific rigor.”
Response 2: Thank you for this helpful comment. We have revised the abstract to improve punctuation consistency and grammatical clarity. We have also moderated phrases such as “central role” to enhance precision. Additionally, we clarified terminology related to therapeutic agents, specifying the type and mechanisms of action of drugs like tirzepatide (i.e., dual incretin agonist). These updates can be found in the revised Abstract section, lines 10-32.
Comment 3: “The associations between obesity, acute/chronic pancreatitis, and pancreatic cancer have been previously discussed in many reviews. This paper may lack sufficient originality unless it emphasizes new therapeutic or molecular pathways, which are not clearly highlighted.”
Response 3: We thank the reviewer for this important observation. In response, we have revised the Introduction and Conclusion to more clearly highlight the unique contributions and scope of this narrative review. Specifically, we emphasize:
- A unified discussion of shared immunometabolic mechanisms linking obesity with AP, CP, and PC.
- A focused appraisal of intrapancreatic fat as a pathophysiological mediator across disease stages.
- Integration of emerging therapeutic strategies, including GLP-1 receptor agonists and dual incretin agents such as tirzepatide, in the context of pancreatic disease prevention and management.
- Contextualization of nutrition and weight loss interventions across disease progression stages, addressing gaps in prior literature.
- Identification of future research directions involving novel biomarkers, omics-based risk stratification, and precision interventions.
These contributions have been explicitly stated in the final paragraph of the Introduction and reinforced in the Conclusion to ensure that the originality and clinical relevance of the work are clear.
Comment 4: “There is no systematic methodology for article inclusion. The authors do not detail: search databases used (e.g., PubMed, Embase); inclusion/exclusion criteria; time period covered; risk of bias assessment. There are no quantitative analyses or novel insights derived from pooled data.”
Response 4: We sincerely thank the reviewer for their valuable feedback. We acknowledge the distinction between systematic and narrative reviews and appreciate the opportunity to clarify our rationale for choosing a narrative review format.
While systematic reviews are indeed essential for synthesizing data from well-defined and homogeneous research questions — particularly in interventional or diagnostic studies — narrative reviews are particularly useful in exploring broad, complex, and evolving topics that span multiple mechanisms, clinical manifestations, and research disciplines. In the present article, our objective was not only to summarize existing studies but also to integrate diverse findings from epidemiology, molecular biology, clinical observations, and therapeutic approaches to provide a comprehensive overview of the multifactorial relationship between obesity and pancreatic diseases, from inflammation to oncogenesis.
Given the breadth of the topic and the heterogeneity of the evidence — ranging from preclinical studies to clinical observations and bariatric outcomes — a narrative approach allowed us to connect insights across different domains and to propose integrative hypotheses that might not be captured by a narrow systematic question. Moreover, narrative reviews remain a valid and valuable scientific tool, especially when the goal is to provide a conceptual framework or stimulate further research in areas where data are still emerging or fragmented.
To improve transparency and rigor, we have now added a Methods section (page 2 – lines 66-83) outlining the databases searched (PubMed), time frame considered, general inclusion/exclusion criteria, and the narrative nature of the synthesis.
We hope this explanation addresses the concern, and we remain open to further suggestions for improving the clarity and usefulness of our work.
We hope this explanation addresses the concern, and we remain open to further suggestions for improving the clarity and usefulness of our work.
Comment 5: “Some therapeutic claims (e.g., anti-cancer effects of orlistat or GLP-1 RAs) are heavily based on preclinical findings, which are presented with insufficient critical appraisal of their translational limits.”
Response 5: We thank the reviewer for their insightful observation. We acknowledge that several of the therapeutic claims—particularly regarding orlistat and GLP-1 receptor agonists—were derived predominantly from preclinical data. To address this concern, we have revised the relevant section to clarify the translational limitations of these findings, distinguish between experimental and clinical evidence, and moderate the language used when describing their potential therapeutic implications. These changes are now reflected in the updated text in Section 4.2 Pharmacologic Interventions (page 10, lines 435-438 and 459-461).
Comment 6: “The discussion of the “obesity paradox” in chronic pancreatitis is not critically dissected, some conflicting data are mentioned, but no mechanistic hypotheses or explanations are elaborated to resolve the paradox.”
Response 6: We thank the reviewer for this insightful comment. We agree that the concept of an “obesity paradox” in CP warrants a more critical and mechanistic appraisal. In response, we have revised the relevant section to expand on potential confounders such as reverse causation, selection bias, and BMI limitations, and we now explore plausible pathophysiological mechanisms — including sarcopenia, fat distribution heterogeneity, and fibrosis-associated compartmentalization of adipose tissue — that may underlie the observed clinical variability. These additions aim to clarify why excess adiposity might be associated with divergent outcomes in CP and to underscore that current evidence does not robustly support a protective effect. Our revisions provide a more nuanced and critically informed discussion to address this complex issue. These updates can be found in the revised section 2.1 Obesity and Risk of Chronic Pancreatitis (pages 5-6, lines 210-225).
Comments 7: “The manuscript provides a comprehensive overview of the relationship between obesity and pancreatic diseases. However, I recommend several improvements to enhance its scientific value. First, the section would benefit from deeper critical analysis of the cited literature, particularly in distinguishing between preclinical and clinical findings, and weighing the strength of evidence. Second, the discussion occasionally overgeneralizes outcomes (e.g., weight loss interventions), without specifying context or patient subgroups. Additionally, while individual disease mechanisms are well presented, greater integration across conditions, such as how obesity may influence the progression from acute to chronic pancreatitis to pancreatic cancer, would strengthen the narrative. The section also lacks an explicit discussion of the limitations of the review itself, including the use of a narrative format and the absence of a systematic methodology. Lastly, several concepts (e.g., lipotoxicity, intrapancreatic fat, GLP-1 therapies) are repeated throughout without deeper synthesis. Addressing these issues would enhance the clarity, depth, and critical rigor of the discussion.”
Response 7: We thank the reviewer for their thorough and constructive feedback, which we believe has helped us substantially improve the clarity, critical depth, and scientific value of our manuscript. In response, we have implemented the following changes:
- Critical appraisal of the literature:
Throughout the revised manuscript, we have strengthened the distinction between preclinical and clinical findings, particularly in Sections 4.2(page 10 lines 435-438 and 459-461) and 4.4 (page 12, lines 542-559). We now more clearly state when therapeutic effects (e.g., of GLP-1 receptor agonists or orlistat and bariatric surgery) are based on animal or in vitro models and critically discuss their translational limitations. We also specify the strength of evidence supporting different mechanisms and clinical associations, using language that better reflects the level of confidence and potential uncertainty. - Clarification of therapeutic effects and generalizations:
In response, we revised the introductory paragraph of Section The Impact of Obesity Treatment on Pancreatic Diseases (page 9, lines 389-393) to explicitly acknowledge the heterogeneity in clinical response to obesity treatments. We now state that the therapeutic impact of interventions—such as lifestyle modification, pharmacologic therapy, endoscopic approaches, and bariatric surgery—may vary significantly depending on patient characteristics (e.g., metabolic profile, comorbidities), disease stage (acute vs. chronic vs. neoplastic), and the type of intervention employed. These clarifications were made to avoid overgeneralization and to reflect the complexity and variability of real-world clinical outcomes. We believe this change addresses the reviewer’s concern and strengthens the applicability and nuance of our discussion. - Improved integration across disease stages:
To better illustrate how obesity may influence the continuum from AP to CP to PC, This integrative perspective is now explicitly addressed at at the beginning of Section 2.2 (page 6, lines 234-243), we now highlight that CP is increasingly recognized as part of a disease continuum originating with episodes of AP. We describe how this evolution involves morphological remodeling of the pancreas, including fibrotic replacement and adipose infiltration—both processes influenced by obesity. At the end of Section 3 Clinical Implications and Future Directions (page 7, lines 276-282), we expand on how chronic inflammation, intrapancreatic fat accumulation, and fibrosis—hallmarks of obesity-associated pancreatic remodeling—may create a permissive microenvironment for acinar-to-ductal metaplasia (ADM), a precursor to pancreatic neoplasia. Together, these additions reinforce the concept of a pathophysiological continuum linking AP, CP, and PC. They also support the emerging hypothesis that obesity-related processes—particularly lipotoxicity and immunometabolic dysregulation—may accelerate this transition. We hope these revisions enhance the conceptual clarity and translational value of the review. - Limitations of the review format:
We have added an explicit discussion of the limitations of our narrative review methodology at the end of the Methods section (page 2). This includes acknowledgment of the lack of systematic article selection and risk of bias assessment, and a justification for the use of a narrative approach, particularly given the breadth and complexity of the topic. - Reduction of repetition and increased synthesis:
We revised sections that previously repeated concepts such as lipotoxicity (section 1.3 – page 4, lines 146-150), intrapancreatic fat (see section 1.2.4 – page 4, line 129), aiming to reduce redundancy and enhance cross-sectional synthesis. For instance, mechanistic insights about intrapancreatic fat now build cumulatively across AP, CP, and PC rather than being reiterated separately.
We trust that these revisions have improved the manuscript’s clarity, coherence, and analytical rigor, and we remain grateful for the reviewer’s helpful comments.
Comment 8: “The should acknowledge the lack of a systematic search strategy, possible selection bias, and the absence of formal quality assessment of the cited studies. Moreover, the reliance on preclinical data and observational studies should be discussed, particularly where mechanistic extrapolations, or therapeutic implications are drawn from animal models or small cohorts.”
Response 8: We thank the reviewer for this important observation. As noted in our response to Comment 4, this review was conceived as a narrative synthesis, given the broad scope and heterogeneity of the topic—which spans mechanistic, preclinical, observational, and interventional data across distinct pancreatic disease stages. Narrative reviews are particularly suitable for integrating diverse forms of evidence and for generating conceptual insights where systematic reviews may be too restrictive.
Nonetheless, we fully agree that transparency and critical appraisal are essential. To address this, we have now explicitly acknowledged the methodological limitations of our narrative approach in the Methods section (page 2), including the absence of a formal systematic search strategy, the potential for selection bias, and the lack of formal quality assessment of included studies.
Furthermore, in Section 4 (pages 9-12), we have incorporated a more critical appraisal of the evidence base, particularly regarding the therapeutic implications of preclinical data and small observational cohorts. Where appropriate, we now explicitly highlight the preliminary or hypothesis-generating nature of such findings and caution against overinterpretation.
We believe these additions increase the rigor and transparency of the manuscript, while preserving the integrative value of the narrative format. We appreciate the reviewer’s suggestion in guiding these improvements.
Comment 9: “The future research directions are mentioned, but they remain somewhat general. The manuscript could be strengthened by proposing more specific research questions or methodologies”
Response 9: We thank the reviewer for this insightful and constructive suggestion. We fully agree that specifying more concrete research questions and methodologies enriches the future outlook and enhances the translational relevance of the review.
In response, we have revised the concluding paragraphs of the manuscript to incorporate several of the reviewer’s proposed directions. Specifically, we now:
- Highlight intrapancreatic fat quantification via imaging as a promising biomarker for progression risk in both AP and CP, with implications for early detection and intervention strategies (see section 2.3 – page 7, lines 283-286).
- Emphasize the need for randomized controlled trials assessing pharmacologic weight-loss therapies (e.g., GLP-1 receptor agonists, dual incretin therapies) on pancreatic cancer progression and outcomes (see Conclusion - page 13, lines 584-593)
- Suggest the investigation of molecular subtypes of obesity-associated pancreatic disease, potentially informed by multi-omic approaches, including lipidomics, transcriptomics, and microbiome profiles (see Conclusion – page 13, lines 594-598)
- Introduce the potential role of AI-based predictive tools and personalized nutrition strategies, particularly in the context of early-stage disease prevention and stratified care (see Conclusion – page 13, lines 598-600).
These additions aim to move beyond general recommendations and provide a more focused and forward-looking research agenda, in alignment with the reviewer’s excellent suggestions.
Comment 10: “The review successfully outlines that weight loss may reduce disease burden; however, this is stated in general terms, without clear clinical guidance.”
Response 10: We thank the reviewer for highlighting the need for clearer clinical guidance regarding obesity-targeting interventions in the context of pancreatic diseases. In response, we have revised the Conclusion section to more explicitly address the clinical implications of our findings.
First, we now emphasize the importance of systematic assessment of obesity and nutritional status in all patients presenting with pancreatic disease (acute, chronic, or neoplastic), recommending its integration into both diagnostic workup and longitudinal care (Conclusion, paragraph 1; section 1.4 – page 4, paragraph 1; section 2.3 – page 6, paragraph 1; section 3.3 – page 8, paragraph 4).
Second, we now provide more detailed guidance on intervention strategies tailored to disease stage and metabolic profile, highlighting the potential benefits of structured dietary counseling and personalized nutritional planning in improving outcomes (Conclusion, paragraph 2).
Third, we acknowledge the therapeutic potential and current limitations of pharmacologic agents, including GLP-1 receptor agonists and tirzepatide, especially in the context of subclinical disease. We explicitly state the need for randomized controlled trials to clarify which patients, at what disease stages, and under what clinical conditions these therapies may be most effective and safe (Conclusion, paragraph 3).
These additions aim to move beyond general statements and provide a more actionable framework for clinical translation. We believe this improves the practical utility of our review while preserving its conceptual scope.
Comment 11: “The conclusion section is generic and doesn’t adequately summarize key clinical implications or concrete future research directions.”
Response 11: We thank the reviewer for this insightful observation regarding the conclusion. In response, we have substantially revised the final section to go beyond a general summary, incorporating specific clinical implications and detailed future research directions.
The updated conclusion now synthesizes the multifactorial role of obesity across the pancreatic disease continuum—highlighting its influence on inflammation, fibrosis, and tumorigenesis—and clearly positions it as both a modifiable risk factor and a therapeutic target. We emphasize that evidence supports benefits of weight-loss strategies not only for metabolic outcomes but potentially for modifying the natural history of pancreatic diseases, depending on intervention type and disease stage.
To address the reviewer’s concern about generality, we also incorporated forward-looking, concrete research priorities. These include studies of multi-omic profiling (e.g., lipidomics, transcriptomics, microbiome) to identify molecular subtypes of obesity-associated pancreatic disease; and the integration of artificial intelligence–based risk prediction tools and personalized nutrition in future prevention strategies. These additions aim to provide actionable, clinically relevant avenues for future investigation.
We believe these modifications substantially enhance the utility and scientific rigor of the conclusion, while offering novel integrative perspectives that respond directly to the reviewer’s feedback.
Response to Comments on the Quality of English Language
We thank the reviewer for their detailed suggestions regarding the English language and style. We have carefully revised the manuscript to address the grammatical and lexical issues raised. Specifically:
- Line 19: “on disease risk and progression” has been corrected to “on the risk and progression of disease”.
- Line 31: “validate biomarkers for personalized risk assessment” has been revised to “identify and validate biomarkers”.
- Line 191: “may even present paradoxical clinical outcomes” has been modified to “may appear to confer paradoxical clinical outcomes”.
- Line 368-369: “Metformin, a cornerstone in T2DM management have shown…” has been corrected to “Metformin, a cornerstone in T2DM management has shown…”.
- Lines 376-377: “exploration not only for cancer prevention but also for their potential…” has been revised to “exploration both for cancer prevention and for their potential…”.
In addition to these specific changes, the entire manuscript has undergone a thorough language revision to improve grammar, style, and clarity. We are confident that the quality of English now meets the journal’s standards, but we remain open to additional revisions should the editorial team consider it necessary.
Round 2
Reviewer 2 Report
Comments and Suggestions for Authors
The authors have significantly revised the manuscript addressing the concern raised. I consider it could be accepted for publication in this journal.
Comments on the Quality of English LanguageI propose to have the manuscript checked by a native English speaking person.